# Effect of Laser Welding Parameters on Joint Structure of AZ31B Magnesium Alloy and 304 Stainless Steel

**DOI:** 10.3390/ma15207114

**Published:** 2022-10-13

**Authors:** Zhe Wu, Jiaqi Wan, Yang Zhang, Bo Xue, Ruizhi Wu, Chunmei Yang

**Affiliations:** 1College of Mechanical and Electrical Engineering, Northeast Forestry University, Harbin 150040, China; 2College of Science, Northeast Forestry University, Harbin 150040, China; 3College of Materials Science and Chemical Engineering, Harbin Engineering University, Harbin 150001, China

**Keywords:** nanosecond laser welding, microstructure, numerical simulation

## Abstract

The effects of laser welding parameters on the interface microstructure of AZ31B magnesium alloy and 304 stainless steel were investigated. After welding, a scanning electron microscope and ultra-depth of field microscope were used to observe the microstructure of the welded material, to analyze the effects of power on the interface morphology. The simulation of laser welding of magnesium and steel was carried out by the COMSOL software. The results showed that when the power was 15 W–20 W, the temperature did not reach the melting point of magnesium alloy, there was MgO at the welding, and the interface had poor connection strength. When the power was 35 W–50 W, the temperature reached or even exceeded the boiling point of magnesium alloy, and the interface formed hot cracks, pores, and oxides and had poor joint strength. When the power was 25 W–30 W, the temperature was between the melting point and boiling point of magnesium, and the interface had excellent connection strength.

## 1. Introduction

In order to save energy and reduce emissions, the automobile gradually developed to miniaturization and lightweight. Therefore, using light metal material in body manufacturing constitutes a higher proportion. A magnesium alloy has many advantages, such as low density, high specific strength, strong shock absorption ability, and good corrosion resistance, which has attracted much attention in the selection of body materials. At the same time, steel is a traditional material in the automobile industry, widely used in body parts. Therefore, the combination of magnesium alloy and steel is essential in the process of vehicle manufacturing. The realization of magnesium alloy and steel welding can effectively reduce the quality of the vehicle, which has great practical significance. However, due to the physical and chemical properties of magnesium and steel, the melting point of iron (1811K) is higher than the boiling point of magnesium (1363K). Therefore, in the steel welding and melting process, magnesium may evaporate directly, which brings great challenges to the welding of magnesium and steel. To date, connections between magnesium and steel have been made, and various welding techniques have been employed [1,2,3,4,5,6,7]. H. Zhou et al. conducted laser fusion welding experiments with aluminum foil in DP590 dual-phase steel and AZ31B magnesium alloy. The results show that adding aluminum foil can effectively avoid the defects such as porosity, cracks, and softening of the heat-affected zone, improving the bonding strength of steel and magnesium joints and effectively adjusting the heat transfer between steel and magnesium under the action of a high-power laser [8]. L. Liu et al. investigated the lap welding of nickel and copper interlayers on AZ31B magnesium alloy and Q235 low carbon steel, adopting laser-tungsten argon arc welding composite welding technology. The results show that the shear strength of the copper joint is slightly higher than that of the nickel joint [9]. M. Wahba et al. studied the lap welding between AZ31B and galvanized steel. The results show that the welding is unstable due to the difference in physical and chemical properties between magnesium and steel. By switching to the transmission mode, the laser beam does not penetrate the material under the action of the liquid metal film of galvanized steel, and hence the stability of welding is improved, and successful welding can be achieved. The presence of zinc coating plays a great role in eliminating the negative effects of oxide on the bonding process [10]. G. Casalino et al. performed laser bias welding of AZ31B and 316 stainless steel without any sandwich or groove preparation. The results showed that rupture was observed on the magnesium side off-center line due to the formation of a hard thin layer of metal compounds in the molten zone. The molten zone has effective metallurgical bonding and no mixing under the liquid. Low-strength welding is proving to be a promising technique for joining dissimilar metals to produce effective bonding with good tensile strength [11]. T. Tao et al. explored a new laser welding connection method between AZ31B magnesium alloy and DP590 dual-phase steel, in which tin powder with grooves was added to the lap joint. The results show that the interface between AZ31B magnesium alloy and DP590 dual-phase steel can be connected, and the welding quality is good. When the width of the groove with tin powder is 1.5 mm, the maximum load is doubled compared to the one without the groove. The meshing effects of the groove and the metallurgical effects of adding tin powder improve the mechanical properties of the magnesium and steel joint [12]. However, there are few reports on welding dissimilar metal thin-walled parts at low strength.

In the current study, it is particularly urgent to weld ultra-thin magnesium alloy and steel. However, there is limited research on the welding of magnesium and steel sheets. Therefore, this paper used low power for sheet 304 stainless steel and AZ31B magnesium alloy to conduct nanosecond pulse laser welding experiments and mechanism research on steel and magnesium under the steel to solve the industry demand. At the same time, the microstructure of welds under different welding powers was observed and analyzed by means of a scanning electron microscope and ultra-depth field microscope. The bonding mechanism of magnesium alloy and steel under different welding powers was discussed. The research results provide a theoretical basis for the welding of magnesium and steel dissimilar metals.

## 2. Materials and Methods

### 2.1. Selected Materials and Preparation Process

In this work, 304 stainless steel (SS304) and AZ31B magnesium alloy were chosen. The sizes of sheet 304 and AZ31B were 20 × 10 × 0.1 mm and 20 × 10 × 0.5 mm, respectively. The steel plates were placed on the magnesium plate in an overlapping form. On the one hand, because the reflectivity of steel was less than that of magnesium, the steel would melt under laser irradiation, and the heat was then transferred to the magnesium plate. On the other hand, in the process of welding, the steel can sink down due to gravity, while the magnesium alloy floats up under the influence of buoyancy in the molten state, which is conducive to the combination of steel and magnesium. Meanwhile, 304 stainless steel produced by Taiyuan Iron and Steel and AZ31B magnesium alloy produced by the Zhenxin Magnesium Company were used in the experiment. The specific composition is shown in Table 1, and some physical properties of stainless steel and AZ31B magnesium alloy at room temperature are shown in Table 2.

Before welding, the 304 stainless steel and AZ31B magnesium alloy should be cleaned by different cleaning processes. Then, the sandpaper of 200#-2000# was selected for successive polishing. After polishing, the gravel on the surface of the sample was washed with water. During polishing, the rotating speed of the polishing disc was adjusted to 300 RPM, and the particle size of a 2.5 μm diamond spray polishing agent was selected. In order to prevent friction between the sample surface and the polishing cloth from the heating, a distilled water coolant with or without ethanol was added to the front of the sample. The sample was polished to the surface and had luster, and no obvious scratch could be observed under a 500X optical microscope. After polishing, the sample was cleaned with anhydrous ethanol for 15 s. Then, dry the sample surface with a hair dryer (cold air) to prevent oxidation of the sample surface.

### 2.2. Laser Welding Process

The experiment laser is a nanosecond pulse laser produced by China Jingwei Laser. The device model is JW-F30W, with a wavelength of 1064 nm, a spot size of 0.01 mm, and a defocus of 0. The laser beam processes the surface through the galvanometer. Table 3 shows the details of the laser parameters. Then, the 304 stainless steel and AZ31B magnesium alloy were fixed on the test table with the pattern of magnesium on steel and magnesium on steel. Because this experiment was welding thin plate dissimilar metal with low power, a concentric circle path with small heat accumulation was selected to process. With the other parameters fixed, the samples were processed by changing different power at room temperature. The ultra-depth microscope used in the experiment is the VHX-2000, produced by Keyence, a Japanese company, with a magnification of 100 to 5000 times. To observe the cross section of the welded surface, a Scanning Electron Microscope (SEM) was used in the experiment, produced by COXEM Of South Korea, and the instrument model was EM-30 Plus. The device has a resolution of 5 nm, a magnification from 20 to 150,000 times, and is operated using the NanoStationTM3.0 software system. The schematic diagram of laser welding is shown in Figure 1.

### 2.3. Numerical Simulation of Temperature Field

COMSOL software was used to simulate the temperature field in the welding process. The model is shown in Figure 2. The welding trajectory is the same as the actual trajectory. The heat source Q used in welding simulation calculation is the plane Gaussian heat source, which is defined as:(1)Q=2Pnπr2×exp−2d2r2
(2)$$d=\sqrt{{\left({x–x}_{\mathrm{laser}}\right)}^{2}+{\left({y–y}_{\mathrm{laser}}\right)}^{2}}$$
where n is the absorption coefficient of the laser beam, p is the laser power, r is the radius of the laser spot, d is the distance from the workpiece to the heat source, $x_{\mathrm{laser}}$ and $y_{\mathrm{laser}}$ are the coordinates of the center of the laser spot.

There are two main heat transfer modes in the workpiece: solid heat transfer and fluid heat transfer. Both convection and radiation occur between the workpiece and the environment. Molten metal flows in the pool for the following three reasons: one is the uneven distribution of surface tension caused by an uneven temperature. The second is the material density change caused by the temperature change. Third, the influence of gravity and buoyancy.

The simulation process of the welding temperature field can be regarded as the process of solving the differential equation of heat transfer under the initial conditions and boundary conditions. The initial temperature T1 of the workpiece is the same as the ambient temperature T0. The heat transfer control equation is as follows:(3)$${\rho C}_{p}\frac{\partial T}{\partial t}{+\rho C}_{p}\mu \cdot \nabla T+\nabla \cdot \left(-k\nabla T\right)=Q$$

ρ is the density of the material, $C_{p}$ is the constant pressure heat capacity of the material, k is the material’s thermal conductivity, and μ is the velocity component. Fluid flow in a molten pool follows the continuity equation and motion equation:(4)$$\rho \frac{\partial u}{\partial t}+\rho \left(\mu \cdot \nabla \right)\mu =\nabla \cdot \left(-\mathrm{pI}+K\right){+F}^{\prime }+\rho g$$
(5)$$\frac{\partial \rho }{\partial t}+\nabla \cdot \left(\rho \mu \right)=0$$
(6)K=μ∇μ+∇μ2−23μ∇·μI
where μ is the dynamic viscosity of the material, ρ is the stress tensor of the fluid, F is the volume force, K is the viscous stress, and I is the strain rate tensor. The welding process with different power parameters from 15 W to 50 W was simulated numerically. The welding parameters were optimized through a series of experiments.

## 3. Results and Discussion

### 3.1. Cross Section and Theoretical Analysis of the Welding Seam

Figure 3 shows the influence of different power on the cross-section of the weld by ultra-depth of field microscope. As shown in Figure 3a, when the power is 15 W, the magnesium alloy is melted due to heat conduction, resulting in a large number of inclusions between the magnesium and steel, with a thickness of about 54.39 μm. Due to the active chemical properties of magnesium, MgO, with a high melting point and high density, is easily formed at high temperatures. MgO is not easily discharged from the alloy solution with low density, resulting in the formation of a flake slag. MgO slag inclusion will reduce the plasticity of weld metal and make the joint brittle [13,14,15]. In the case of low power, magnesium and steel cannot easily form good welded joints. As shown in Figure 3b, when the power is 20 W, a small amount of magnesium is bound to the steel side under the influence of buoyancy in the molten state. The MgO slag inclusion layer (36.84 μm) between magnesium and steel becomes less. Formula (7) is the relationship between the density and temperature of magnesium alloy liquid metal:(7)$${\rho =\rho }_{m}+\frac{\partial \rho }{\partial T}\left({T-T}_{m}\right)$$
where, $\rho_{m}$ is the density of the AZ31B magnesium alloy at the melting point temperature ($\rho_{m}=1.59\times 10^{3}\;\mathrm{kg}\cdot m^{-3}$), $\frac{\partial \rho }{\partial T}$ is the density temperature variation coefficient of the AZ31B magnesium alloy (−0.26 kg·m^−1^·K^−1^), and $T_{m}$ is the melting point of the AZ31B alloy (923.15 k). Formula (7) shows that the density of the liquid metal of magnesium alloy decreases with the increase in temperature, which makes the fluid mass per unit volume near the edge of the molten pool larger than that at the center of the surface. Therefore, the fluid at the edge flows down along the boundary of the molten pool and flows up at the center under the action of gravity. At this time, the density of steel in the molten state ($\rho_{m}=7.7\times 10^{3}\;\mathrm{kg}\cdot m^{-3}$) is greater than that of magnesium and flows downward under the action of gravity [16]. As shown in Figure 3c, when the laser welding power is 25 W, magnesium and steel are fully bonded with a small amount of oxide produced on the magnesium side. The depth of magnesium alloy bonding to the steel side increases due to buoyancy. As shown in Figure 3d,e, when the power is 30 W and 35 W, there are fewer oxides. The welding surface of magnesium and steel is well bonded, and the strength of the joint increases. As shown in Figure 3f, as the power increases, the penetration depth of magnesium into the steel side increases, and the strength of the joint decreases under laser scanning when the steel enters the molten state. As shown in Figure 3g,h, since the welding method is to adopt a concentric circle from outside to inside, when the welding starts, the excessive power leads to the heat conduction of the magnesium alloy at the center to become a molten state, and a large number of oxides are formed in the outer magnesium alloy. The energy distribution of a pulsed laser beam can be approximated as Gaussian distribution, and the temperature distribution of liquid metal on the molten pool surface (T) is proportional to the energy distribution of the laser beam (q):(8)q=3AρEexp−3r2r02
where, A is the laser absorption coefficient (%), $r_{0}$ is the spot radius (mm), and r is the distance from the beam center (mm). Thus, the smaller the distance (r) is to the center of the molten pool surface, the more heat (q) is absorbed and the higher the temperature (T) is. When laser scanning gradually moved toward the center, magnesium alloy began to evaporate under the influence of heat accumulation and formed large pores [17,18].

During the nanosecond pulse laser treatment, most of the laser energy is absorbed by the surface of the material, resulting in a significant increase in the temperature and the formation of a small melt pool. On the other hand, when the temperature exceeds the boiling temperature, a small amount of melt on the surface rapidly evaporates. According to the principle of thermodynamics, the size of the welding heat input will directly affect the peak temperature, and the relationship between peak temperature and heat input is as follows:(9)Tmax=T0+0.234Ecρr02

In the formula, $T_{\max}$ and $T_{0}$ respectively represent the peak temperature and initial temperature of the weldment, $c_{\rho }$ is the volume-specific heat capacity of the material to be welded, $r_{0}$ is the linear distance from a point of the weldment to the laser focus, and is the welding heat input. According to Formula (9), as the welding heat input increases, the peak temperature of the weldment increases, and the energy obtained by the weld microstructure grain increases [19]. With the continuous increase in power, magnesium alloy penetrates deeper and deeper into the steel side, as shown in Figure 4a, and the melting width of the laser scanning zone becomes larger and larger, as shown in Figure 4b. From top to bottom, each broken line is at the center point, 0.2 mm, 0.4 mm, 0.6 mm, 0.8 mm, 1 mm, and 1.2 mm away from the center point.

### 3.2. Microstructure Analysis of Welds

In order to further observe the microscopic state of the weld, the cross-section of the weld was observed by SEM. In Figure 5a, a small amount of oxide appeared in the thermal reaction zone of the laser welding zone. In Figure 5b, there is oxide slag inclusion in magnesium and steel, and MgO and arc thermal cracks appear in the laser scanning area of the steel side. This kind of crack is caused by the transformation of the stress of molten metal from a compressive state to a tensile state. During the cooling process of the molten pool, the thermal expansion coefficients between magnesium and steel are quite different. These differences further lead to the formation and expansion of thermal cracks. In Figure 5c, it is found that there are relatively few oxide interlayers between steel and magnesium, and there are a few thermal cracks in the laser scanning region. In Figure 5d, it is found that magnesium begins to invade the steel side in the molten state, while a small amount of steel is retained in magnesium. Figure 5e,f shows the element distribution of P1 and P2 analyzed by an EDS dot plot. The scanning results show that the contents of Mg and O elements in the oxides on the magnesium side and between magnesium and steel are high, while the contents of Fe, C, and Cr elements are low. This indicates that the oxides of p1 and p2 are MgO, which also indicates that most of the oxides produced by welding magnesium and steel are MgO. Figure 6 shows that the density of magnesium is lower than that of steel in the molten state, which leads to the phenomenon that magnesium floats upward and steel sinks downward.

Figure 7a shows that there are few oxides between magnesium and steel, and small pores appear in the heat-affected zone. The depth of magnesium invading steel is small, and the welding interface between magnesium and steel is well bonded. It is found in Figure 7b that the welding state of magnesium and steel is good, and there is no crack in the laser welding zone. The shape of magnesium intrusion steel is also mountain-shaped. In Figure 7c, there is a liquefied crack in the heat-affected zone of welding. The liquefaction crack in the heat-affected zone is the result of the interaction between metallurgical and mechanical factors and is also related to tensile stress. Under the action of welding heat, the plasticity and strength of magnesium alloy with a low melting point in the heat-affected zone decrease sharply. Liquefaction cracks are formed under tensile stress. Due to the use of nanosecond pulse laser welding, the cooling speed is fast. At the end of welding, the molten pool and the heat-affected zone cooled rapidly, and the heat-affected zone produced large tensile stress, which was the direct factor for the initiation and propagation of liquefaction cracks in the heat-affected zone during the cooling process. [20,21]. In Figure 7d, there are some oxides in the thermal reaction zone, and the shape of magnesium invading the steel is elliptical. In Figure 7e, it can be found that a small amount of steel residue sinks along the direction of the thermal reaction zone; because the laser power increases, the laser can quickly penetrate the steel layer into the magnesium layer. At this time, the molten magnesium can quickly invade the steel side, which can be obviously observed that the depth of magnesium intrusion into the steel increases with a small amount of pore. This is because magnesium alloy weld has a high sensitivity to produce pores. The gas-producing pores in magnesium alloy weld metal come from hydrogen, mainly from the gas, surface moisture, organic matter, and air inside the welding material AZ31B. Moreover, when the aluminum content of AZ31B is low, the weld is more likely to produce pores because the solid-liquid temperature range is small, which is not conducive to the floating of bubbles. When the welding heat input is small, the oxide film near the groove is not completely melted, so the water in the oxide film is heated to decompose hydrogen. Because the bubbles are generated on the residual oxide film, it is not easy to escape from the floating, resulting in the formation of pores. In addition, the cooling rate and crystallization rate of the welding pool will increase without the protective gas at low power, which is also the reason for the formation of pores. It is found in Figure 7f that the stress change on the steel side results in cracks due to the invasion of magnesium [22].

In Figure 8a,d, as the power increases, it can be found that in the laser welding area far from the center point, the depth of magnesium invading the steel increases, and there is magnesium that has been completely filled into the steel along the laser welding area. There are lots of oxides near the center. It is found in Figure 8e,f that an explosion hole is generated at the welding center at the power of 50 W. At the beginning of the laser scanning, the magnesium alloy at the center was melted due to the influence of heat conduction. When the laser scanning is at the center, the magnesium alloy has reached the boiling state. In this case, the magnesium alloy at this location continues to be affected by heat accumulation, resulting in partial combustion loss and gasification. This forms an explosion hole [23,24,25,26].

Figure 9 shows the optical three-dimensional surface morphology under different welding power. As the power decreases, the laser scanning trajectory becomes clearer. The three-dimensional color contrast shows that the surface tends to be flatter with the decrease in power. When the power is 15 W, the surface is the clearest, and there is no obvious protruding oxide. When the power is 20 W−30 W, there are fewer oxide protrusions on the surface. When the power is 35 W–50 W, the area of the melting pool and paste zone is expanded due to a large amount of heat accumulation. With the increase of power, the laser scanning trajectory gradually blends, and the surface oxide protrusion height gradually increases, as shown in Figure 10.

### 3.3. Welding Temperature Field Simulation

Figure 11 shows the influence of different power on the welding temperature field distribution at 1.8431 s after the start of welding. It was found that with the continuous improvement of power, the temperature of the same section would gradually increase. Figure 11a shows the temperature field simulation of 15 W power, at which the maximum temperature is (670K), and the welding method is heat conduction. At this time, the temperature has not reached the melting point of magnesium alloy, resulting in a poor welding effect. When the power is between 20 W and 45 W, the temperature increases, and the range of heat conduction becomes larger. The highest temperature on the magnesium plate is always between the melting point and boiling point of magnesium. Therefore, the magnesium plate has sufficient melting depth, and the magnesium plate will not vaporize. Figure 11h shows the temperature field simulation of 50 W, the maximum temperature (1360K) has exceeded the boiling point of magnesium alloy, and the magnesium plate began to vaporize. With the continuous growth of welding time and the increase of heat accumulation, the magnesium plate is severely vaporized, resulting in the explosion hole phenomenon at the welding place [27].

At 2.087 s after welding, the phase transition in the cross-section of the vertical welding direction is shown in Figure 12. At this point, the temperature on the magnesium plate has exceeded the boiling point of magnesium. During the welding process, the magnesium plate will vaporize violently, forming pores in the welded joint, and the surface of the weldment will collapse seriously. Since it takes a certain time for the laser energy to transfer from the surface to the inside of the steel plate, the formation time of the molten pool at the bottom of the workpiece is later than that at the top, such that the size of the simulated molten pool and the actual molten pool cannot be directly compared. However, the simulation results of the maximum width of the molten pool and the maximum width of the heat-affected zone on the steel side are equivalent to the actual size. Therefore, the simulation model is reasonable for calculating and analyzing the temperature field of steel and magnesium laser welding [28,29,30,31,32,33,34,35,36].

## 4. Conclusions

(1)When the laser power is 15 W and 20 W, because of the low laser power, MgO slag inclusion exists between magnesium and steel, and hot cracks appear on the steel side. The welding effects of magnesium and steel are poor. From the distribution of the temperature field, we observe this in the same section at 1.8431 s after welding. At 15 W-20 W, the temperature did not reach the melting point of magnesium alloy, resulting in poor adhesion between the steel plate and the magnesium plate.(2)When the laser power is 35 W–50 W, due to the high temperature, the depth of the magnesium intrusion into the steel side increases, resulting in poor joint strength. The depth of the thermal reaction zone becomes deeper, resulting in the formation of thermal cracks, pores, and oxides. Meanwhile, laser welding near the center will form an explosion hole and thermal crack defects, resulting in poor strength of the joint.(3)When the power is 25 W and 30 W, the temperature is between the melting point and boiling point of the magnesium. The depth of the magnesium intrusion to the steel side is appropriate, and there is no MgO inclusion at the joint between magnesium and steel, which has good joint strength and good welding interface bonding.

## Figures and Tables

**Figure 1 materials-15-07114-f001:**
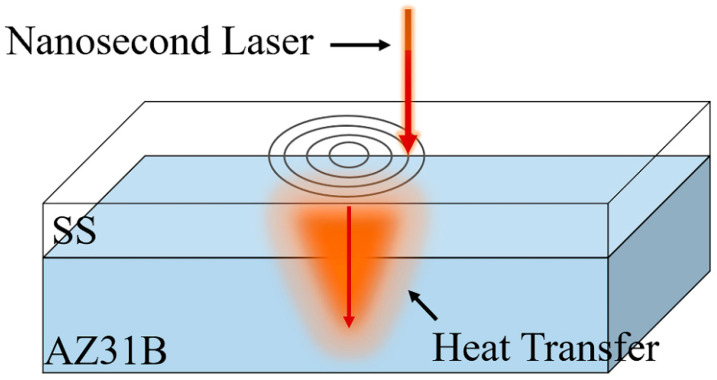
Schematic diagram of the laser welding.

**Figure 2 materials-15-07114-f002:**
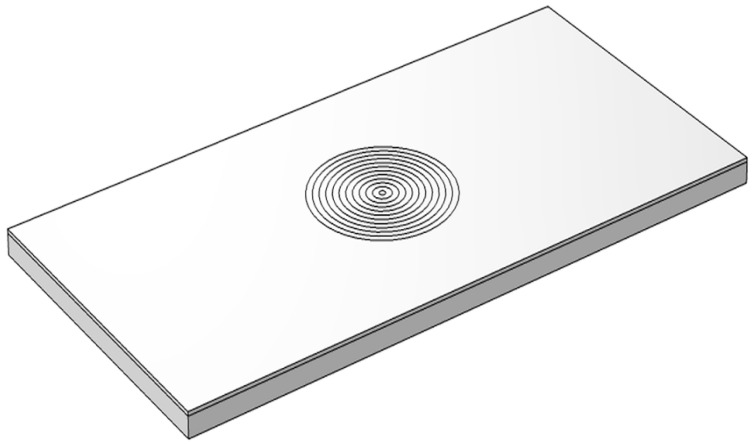
Simulation model.

**Figure 3 materials-15-07114-f003:**
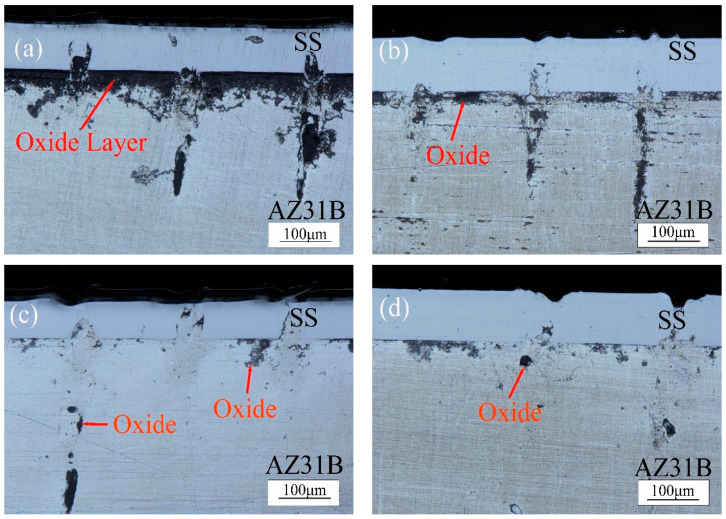
Cross section states of the weld seams at different powers (**a**) 15 W, (**b**) 20 W, (**c**) 25 W, (**d**) 30 W, (**e**) 35 W, (**f**) 40 W, (**g**) 45 W, (**h**) 50 W.

**Figure 4 materials-15-07114-f004:**
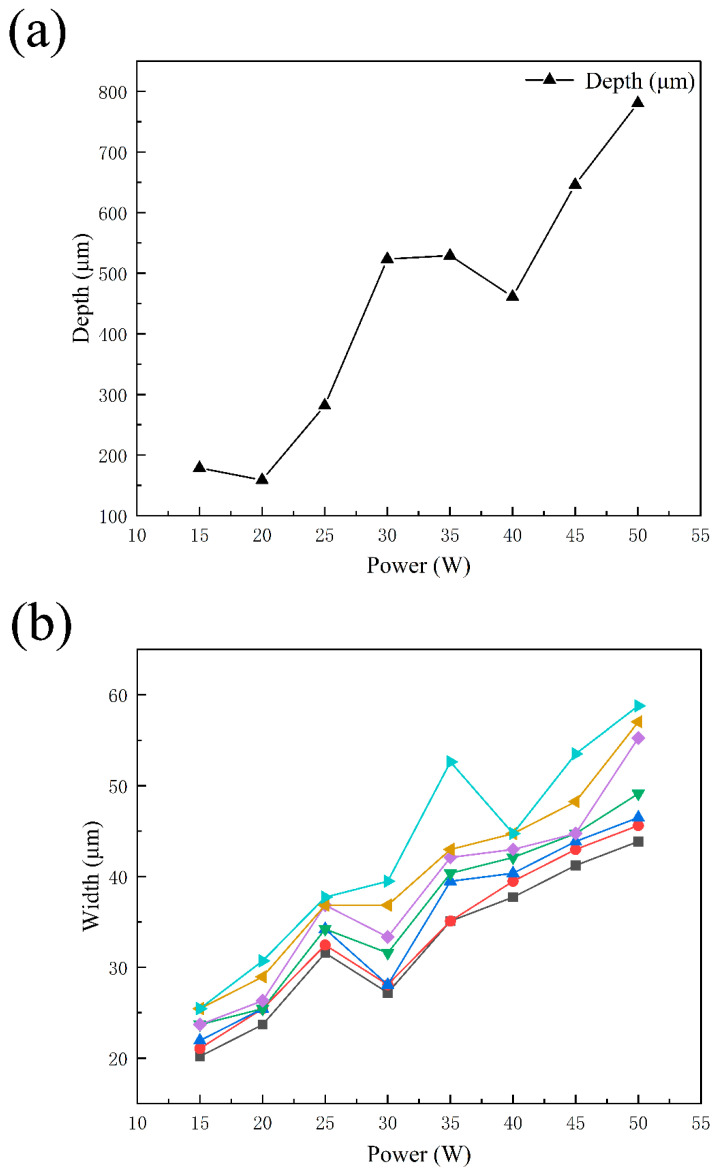
(**a**) Average penetration depth of intruded steel, (**b**) penetration width of laser scanning zone.

**Figure 5 materials-15-07114-f005:**
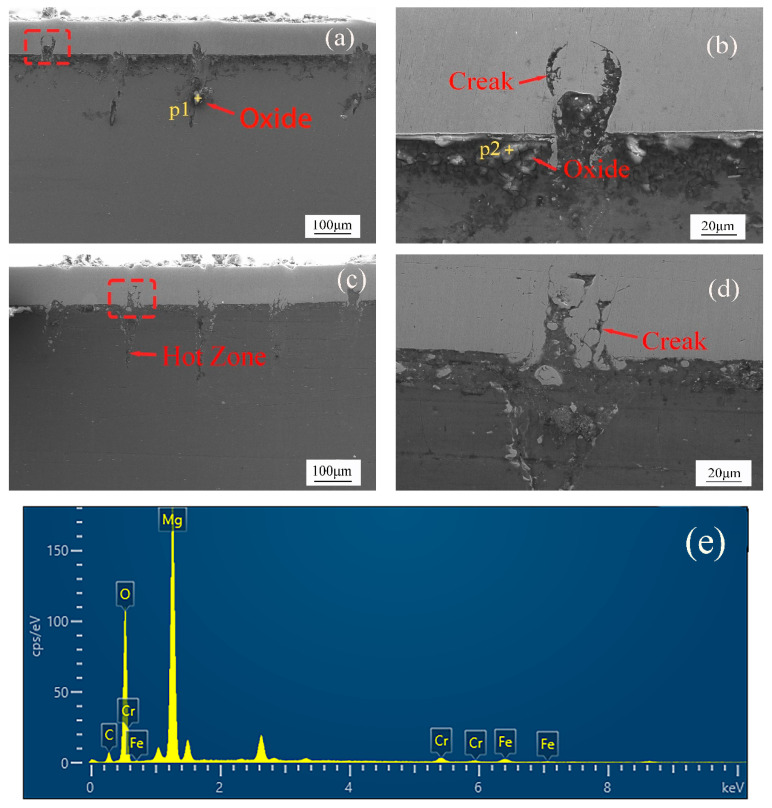
Microstructure and interfacial element distribution of the welded joints: (**a**,**b**) laser power 15 W, (**c**,**d**) laser power 20 W, (**e**) element distribution at p1, (**f**) element distribution at p2.

**Figure 6 materials-15-07114-f006:**
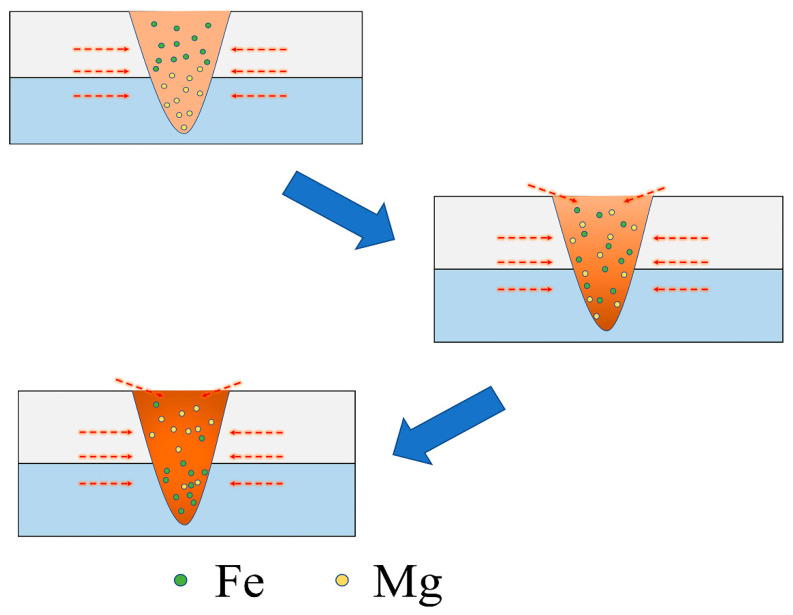
Flow chart of the prime diffusion.

**Figure 7 materials-15-07114-f007:**
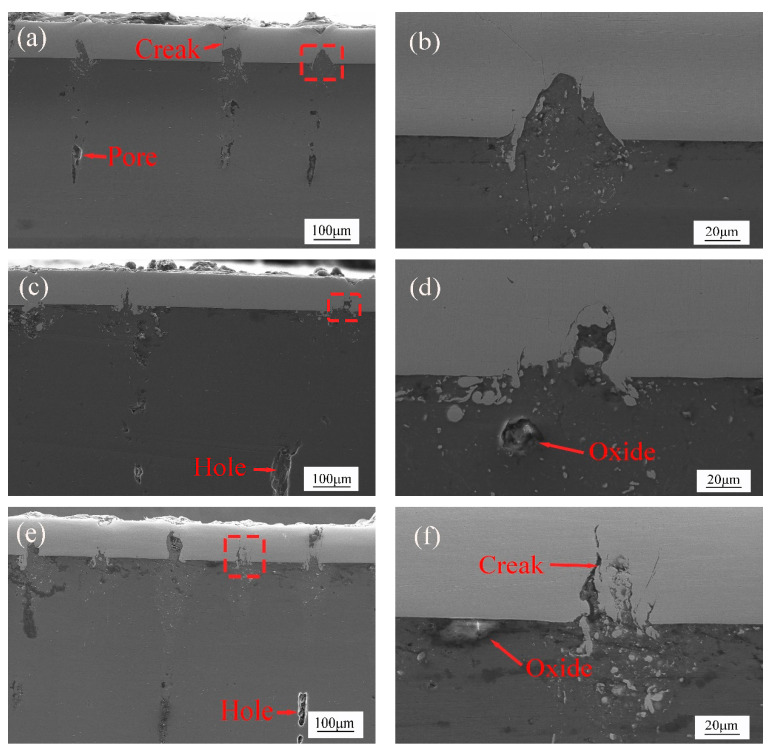
Cross sections of the welding seams under scanning electron microscopy (**a**,**b**) laser power 25 W, (**c**,**d**) laser power 30 W, (**e**,**f**) laser power 35 W.

**Figure 8 materials-15-07114-f008:**
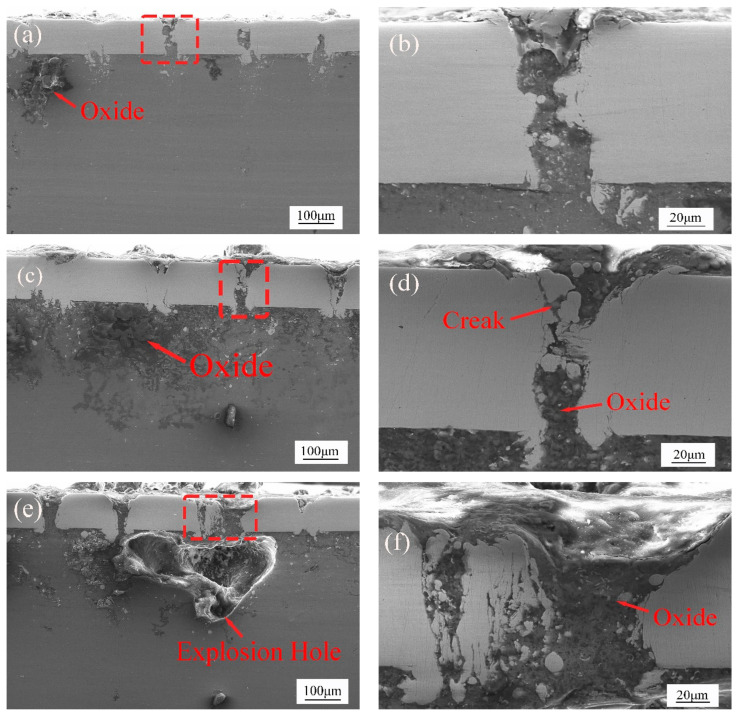
Cross section of weld (**a**,**b**) laser power 40 W, (**c**,**d**) laser power 45 W, (**e**,**f**) laser power 50 W.

**Figure 9 materials-15-07114-f009:**
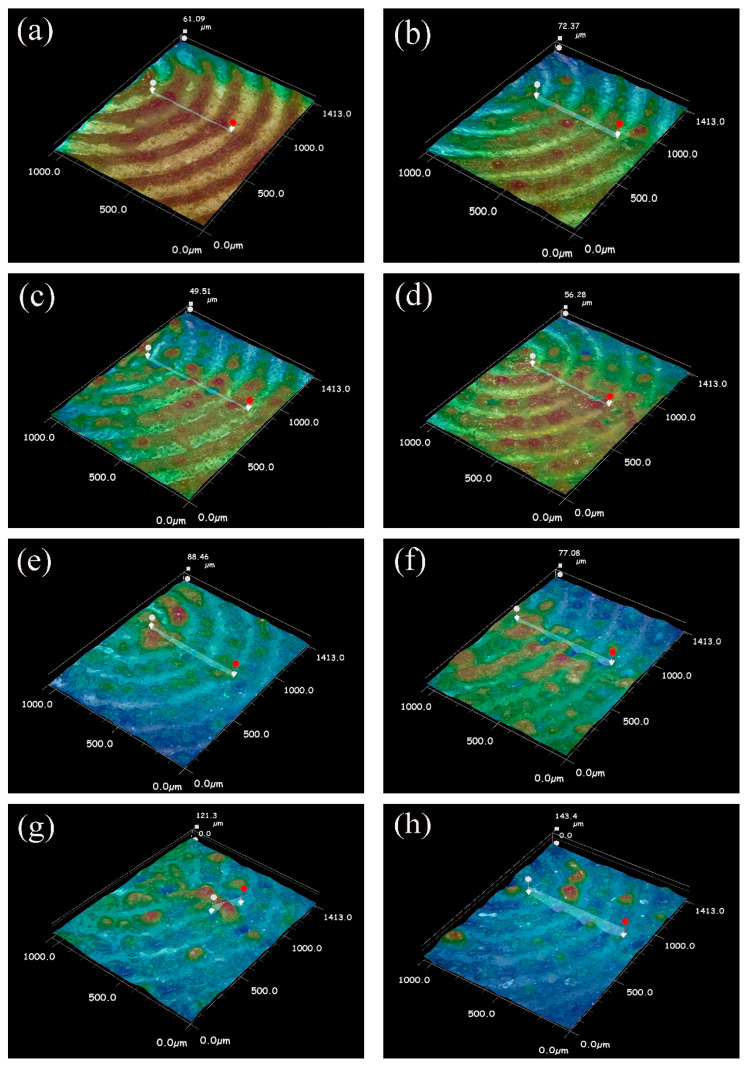
Optical three-dimensional surface morphology under different power (**a**) 15 W, (**b**) 20 W, (**c**) 25 W, (**d**) 30 W, (**e**) 35 W, (**f**) 40 W, (**g**) 45 W, (**h**) 50 W.

**Figure 10 materials-15-07114-f010:**
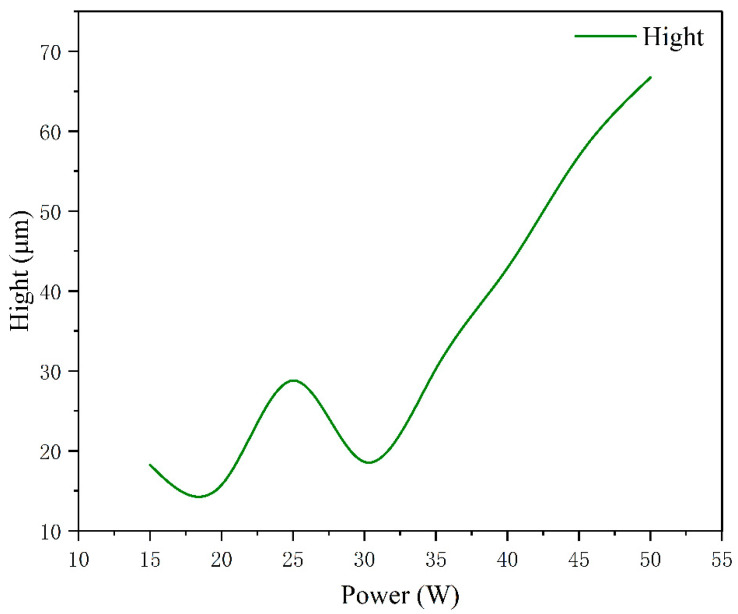
Oxide heights of welded surfaces at different powers.

**Figure 11 materials-15-07114-f011:**
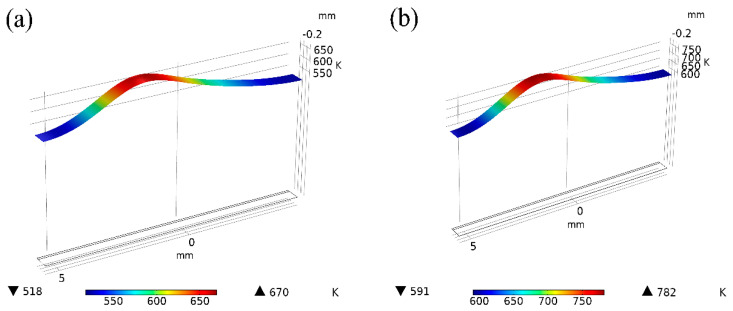
Temperature field of different power under the same section (**a**) 15 W, (**b**) 20 W, (**c**) 25 W, (**d**) 30 W, (**e**) 35 W, (**f**) 40 W, (**g**) 45 W, (**h**) 50 W.

**Figure 12 materials-15-07114-f012:**
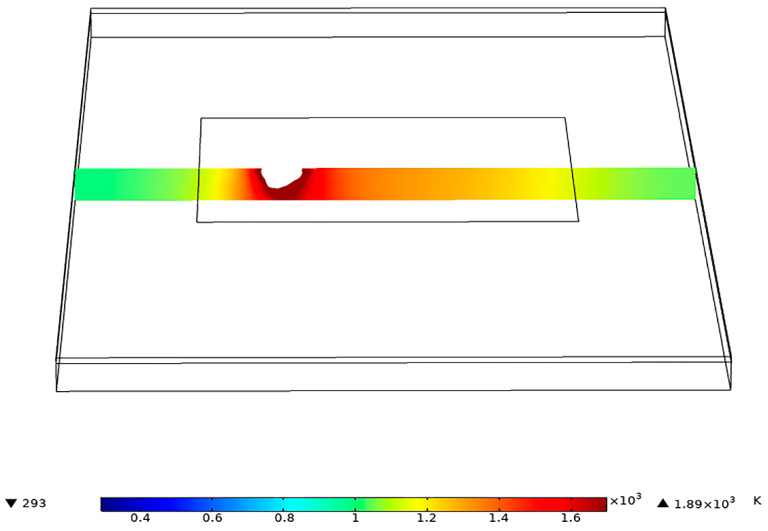
50 W phase transition.

**Table 1 materials-15-07114-t001:** Nominal chemical composition of the 304 stainless steel and AZ31B (wt%).

Sample	Al	C	Si	Mn	P	Cr	Ni	Mg	Fe
304SS	-	0.021	0.386	1.462	0.031	18.137	8.064	-	Bal.
AZ31B	3.00	-	0.02	0.31	-	-	-	Bal.	0.005

**Table 2 materials-15-07114-t002:** Part physical properties of the 304 stainless steel and AZ31B.

Metal	Melting Point (K)	Coefficient of Thermal Conductivity (W·m^−1^·k^−1^)	Density (kg·m^−3^)
304SS	1671.15~1727.15	21.5	7850
AZ31B	923.15	155.5	1738

**Table 3 materials-15-07114-t003:** Nanosecond pulse laser welding parameters.

Welding Parameters	Value
Welding speed (mm/s)	30
Frequency (kHz)	20
Sweep line spacing (mm)	0.2
The pulse width (ns)	10
Power (W)	15, 20, 25, 30, 35, 40, 45, 50

## Data Availability

The data presented in this study are available from the corresponding authors upon reasonable request.

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
