# Peer review of "Effect of Laser Welding Parameters on Joint Structure of AZ31B Magnesium Alloy and 304 Stainless Steel"

_materials, 2022, doi:10.3390/ma15207114_

Round 1
Reviewer 1 Report
1. Authors mentioned that their main novelty of this work was to weld ultra-thin Mg and steel. What is the application of these ultra-thin welded sheets in Automobile industries?
2. How the authors had selected the thickness of sheets? As this will affect the quality of the weld.
3. What material model was used to define the sheet in the simulation?
4. Authors shall include few figures of welded sheet, as it would benefit the readers. Moreover it will be better if any mechanical testing is done to justify their results.
5. How the authors have identified the type of oxides formed in weld by SEM analysis?
6. Conclusion of the article can be succinct. Authors can list their important findings.
Author Response
Dear Reviewers:
Thanks for the reviewers' comments concerning our manuscript entitled“Effect of laser welding parameters on joint structure of AZ31B magnesium alloy and 304 stainless steel”(ID: 1951404).Those comments are valuable for revising and improving our paper with important guiding significance. We have made correction according to the comments, revised portion are marked in red in the paper. The responds to the reviewer's comments are as follows:
Point 1: Authors mentioned that their main novelty of this work was to weld ultra-thin Mg and steel. What is the application of these ultra-thin welded sheets in Automobile industries?
Response 1: We appreciate your advice. In order to be more clear and in line with the reviewer's concerns, we have added a brief explanation as follows: Many studies now try to weld aluminum and steel. Magnesium alloy is the metal material with the lowest density among commonly used metals. The density of magnesium is 2/3 of that of aluminum and 1/4 of that of iron. The research, development and utilization of magnesium alloy can promote the development of lightweight technology. Therefore, this paper tries to use magnesium alloy instead of aluminum alloy to apply the battery case of electric vehicle (EV) containing a large number of batteries. Here are the references:
1)Coroado, J., Ganguly, S., Williams, S. et al. Comparison of continuous and pulsed wave lasers in keyhole welding of stainless-steel to aluminium. Int J Adv Manuf Technol 119, 367–387 (2022). https://doi.org/10.1007/s00170-021-08226-5
2)M.F.R. Zwicker, M. Moghadam, W. Zhang, C.V. Nielsen, Automotive battery pack manufacturing – a review of battery to tab joining, Journal of Advanced Joining Processes, Volume 1,2020,100017, ISSN 2666-3309,https://doi.org/10.1016/j.jajp.2020.100017.
Point 2: How the authors had selected the thickness of sheets? As this will affect the quality of the weld.
Response 2: Thanks to the reviewer's suggestions, we have a certain understanding of the thickness range of nanosecond pulse welding sheet metal through literature review, and we know that the thickness range of nanosecond pulse laser welding sheet metal is about 0.02mm-0.5mm, and considering the problem of cost performance, we finally adopt the thickness of the sheet metal used in this paper. Here are the references:
1)Fengyuan Shu, Sicheng Niu, Baohua Zhu, Laijun Wu, Hongbo Xia, Bo Chen, Junming Zhao, Caiwang Tan, Influence of scan line spacing on nanosecond pulse laser welding of 6063 Al to steel thin sheets, Optics & Laser Technology, Volume 145,2022,107497,ISSN 0030-3992,
https://doi.org/10.1016/j.optlastec.2021.107497.
2)Ruining Huang, Xuehao Huang, Dandan Wang, Lijun Yang,
Effect of Swing-Spiral-Trajectory on pulsed fiber laser welding stainless steel/Copper dissimilar metals, Optics & Laser Technology, Volume 156,2022,108516,ISSN 0030-3992,
https://doi.org/10.1016/j.optlastec.2022.108516.
Point 3: What material model was used to define the sheet in the simulation?
Response 3: Thanks for the reviewer's suggestion, our simulation plate uses 304 stainless steel and AZ31B model in the COMSOL simulation software library, which matches the material properties of the experiment.
Point 4: Authors shall include few figures of welded sheet, as it would benefit the readers. Moreover it will be better if any mechanical testing is done to justify their results.
Response 4: Taking into account the reviewer's suggestions, we added a brief explanation as follows: For the problem of the duplication between the ultra-depth of field microscope and the SEM part of the picture, because it is considered to make it easier for readers to find the area where the high multiple SEM picture belongs. In addition, the mechanical test of ultra-thin metal is difficult, and it is currently being done. The mechanical test will be shown in the next article.
Point 5: How the authors have identified the type of oxides formed in weld by SEM analysis?
Response 5: Thanks for your suggestion, we have rewritten this section according to the reviewer's suggestion and added EDS point mapping pictures in FIG. 5 on page 9, which demonstrate the oxide formation and the variable chemical composition of the joint.
Point 6: Conclusion of the article can be succinct. Authors can list their important findings.
Response 6: Thank you for highlighting this deficiency. According to the reviewer's suggestion, we have modified the conclusion on page 16-17.
Please see the revised article in the attachment,Special thanks for your comments.

Reviewer 2 Report
The work is interesting, but requires the expansion of the scope of research. The presented interpretations of the phenomena occurring during the laser welding process require confirmation or reference to research contained in scientific publications. The following are notes:
In the theoretical part, the authors suggest the use of the studied method in connecting elements in the automotive industry. Hence, the choice of 304L steel is strange and not, for example, DP, CP steel. TRIP or BH. In the case of steels characterized by the austenite phase transition structure, the obtained joint could have a completely different structure than the parent material.
How was the depth and width determined in Figure 4? Are these maximum or average values?
SEM observations do not bring anything new compared to optical microscopy. In this case, the analysis of the chemical composition in micro-regions (EDS) is missing. The obtained results could confirm the formation of oxides as well as the variable chemical composition of the joint.
The chapter title "Microstructure analysis of welds" seems wrong. The chapter does not present the microstructural structure of the joint and the heat-affected zone. It seems that the title should be called "joint structure analysis".
The diagram shown in Figure 6 is very interesting. However, questions should be asked on what basis did the authors come to such conclusions? It would be worth confirming the presented theses by analyzing the microstructure of the joint and analyzing the distribution of elements by means of EDS.
Where did the authors observe eutectic in the photos?
What does Figure 9 show? Is this the joint surface?
Figure 11 is not legible in the presented form.
Author Response
Dear Reviewers:
Thanks for the reviewers' comments concerning our manuscript entitled“Effect of laser welding parameters on joint structure of AZ31B magnesium alloy and 304 stainless steel”(ID: 1951404).Those comments are valuable for revising and improving our paper with important guiding significance. We have made correction according to the comments, revised portion are marked in red in the paper. The responds to the reviewer's comments are as follows:
Point 1: In the theoretical part, the authors suggest the use of the studied method in connecting elements in the automotive industry. Hence, the choice of 304L steel is strange and not, for example, DP, CP steel. TRIP or BH. In the case of steels characterized by the austenite phase transition structure, the obtained joint could have a completely different structure than the parent material.
Response 1: We appreciate your advice. In order to be clearer and in line with the reviewer's concerns, we have added a brief explanation as follows: Many studies now try to weld aluminum and 304 stainless steel. Magnesium alloy is the metal material with the lowest density among commonly used metals. The density of magnesium is 2/3 of that of aluminum and 1/4 of that of iron. The research, development and utilization of magnesium alloy can promote the development of lightweight technology. Therefore, this paper tries to use magnesium alloy instead of aluminum alloy to apply the battery case of electric vehicle (EV) containing a large number of batteries. Here are the references:
1)Coroado, J., Ganguly, S., Williams, S. et al. Comparison of continuous and pulsed wave lasers in keyhole welding of stainless-steel to aluminium. Int J Adv Manuf Technol 119, 367–387 (2022). https://doi.org/10.1007/s00170-021-08226-5
2)M.F.R. Zwicker, M. Moghadam, W. Zhang, C.V. Nielsen, Automotive battery pack manufacturing – a review of battery to tab joining, Journal of Advanced Joining Processes, Volume 1,2020,100017, ISSN 2666-3309,https://doi.org/10.1016/j.jajp.2020.100017.
Point 2: How was the depth and width determined in Figure 4? Are these maximum or average values?
Response 2: Thanks for the question raised by the reviewer. Our depth data is measured by the ultra-depth-of-field microscope, and the width is measured by SEM. The value is the average value.
Point 3: SEM observations do not bring anything new compared to optical microscopy. In this case, the analysis of the chemical composition in micro-regions (EDS) is missing. The obtained results could confirm the formation of oxides as well as the variable chemical composition of the joint.
Response 3: Thank you for highlighting this deficiency, and we have rewritten this section as suggested by the reviewer to supplement EDS point mapping pictures in Figure 5 on page 9, which demonstrate oxide formation and variable chemical composition of joints.
Point 4: The chapter title "Microstructure analysis of welds" seems wrong. The chapter does not present the microstructural structure of the joint and the heat-affected zone. It seems that the title should be called "joint structure analysis".
Response 4: Thank you for the title suggested. The precedent version of the title has been replaced, becoming “Effect of laser welding parameters on joint structure of AZ31B magnesium alloy and 304 stainless steel”.
Point 5: The diagram shown in Figure 6 is very interesting. However, questions should be asked on what basis did the authors come to such conclusions? It would be worth confirming the presented theses by analyzing the microstructure of the joint and analyzing the distribution of elements by means of EDS.
Response 5: Thank you for your advice. We added the analysis of EDS element distribution and combined with the analysis of joint microstructure to obtain the situation shown in FIG. 6.
Point 6: Where did the authors observe eutectic in the photos?
Response 6: Thank you for emphasizing this issue. The eutectic mentioned in the article is to explain the cause of thermal cracks, which is not shown in the figure. You have been misunderstood here, and we have changed the content on page 11, lines 6-14.
Point 7: What does Figure 9 show? Is this the joint surface?
Response 7: Thank you for your suggestion. Figure 9 shows the laser machined surface. As shown in Figures 1 and 2.
Point 8: Figure 11 is not legible in the presented form.
Response 8: Thank you for highlighting this flaw. According to the reviewer's suggestion, the unclear problem in Figure 11 on page 14-15 was modified.
Please see the revised article in the attachment,Special thanks for your comments.

Round 2
Reviewer 2 Report
Thank you for taking into account the comments in the review